# Anxious Thoughts and Well-Being in University Students with Anxiety in the Two Years After the COVID-19 Pandemic: The Mediational Relationship of Physical Symptoms and Sadness Rumination

**DOI:** 10.3390/bs14111109

**Published:** 2024-11-18

**Authors:** Elena Ioana Fratea, Manuela Mihaela Ciucurel, Geanina Cucu Ciuhan

**Affiliations:** Department of Psychology, Communication Sciences and Social Work, Faculty of Educational Sciences, Social Sciences and Psychology, University Center Pitesti, National University of Science and Technology POLITEHNICA Bucharest, 110040 Pitesti, Romania; elena_ioana.fratea@upb.ro (E.I.F.); manuela.ciucurel@upb.ro (M.M.C.)

**Keywords:** anxious thoughts, physical symptoms of anxiety, rumination, well-being

## Abstract

Objective: This study examines the relationship between anxious thoughts and well-being, with physical symptoms and sadness rumination as mediators, in young people who suffer from anxiety in the first two years after the COVID-19 pandemic. Methods: A community sample of 198 participants, 94 males and 104 females, aged between 19 and 35 years, all of them experiencing an anxiety disorder in their past, answered an online survey during the years 2022–2023. The instruments were the Rumination of Sadness and Anger Questionnaire, The Burns Inventory, and Ryff’s Psychological Well-being Scale. The data analysis used hierarchical regression. Results: The results show that the conditional indirect effects of anxious thoughts on well-being are statistically significant (*β* = −0.29, *SE* = 0.08, *p* < 0.001) for high physical symptoms of anxiety (*β* = 0.25, *SE* = 0.11, *p* < 0.001) and for high sadness rumination (*β* = −0.82, *SE* = 0.04, *p* < 0.001). Physical symptoms of anxiety (*β* = 0.25, *SE* = 0.11, *p* < 0.001) and sadness rumination (*β* = 0.05, *SE* = 0.07, *p* < 0.001) have a partial serially mediating effect on the relationship between anxious thoughts and well-being (*β* = −0.74, *SE* = 0.02, *p* < 0.001).

## 1. Introduction

Anxiety is ranked by the World Health Organization as the second largest contributor to global disability, with a prevalence of 3.6%, and with huge consequences in terms of lost health [1].

The physical symptoms of anxiety have the role of increasing attention and preparing the body for immediate response [2]. For example, in heart-focused anxiety, the individual experiences intense cardiac symptoms, a high level of attention focused on the heart and monitoring of related stimuli, avoidance behavior (e.g., avoids physical effort) and fear and worry related to heart functioning [3].

Classic cognitive behavioral (CBT) therapy models emphasize the basic idea that “the way you think affects the way you feel” [4]. The cognitive model of anxiety centers on the idea of vulnerability, meaning that the individual perceives himself as subject to danger with a lack of control, and dysfunctional cognitive processes can magnify this sense of vulnerability [5].

Anxious thoughts include worry (the fear of future events) and rumination (the cognitive representation of past stressful events) [6]. These are important components of anxiety because the thoughts and self-preoccupation are so hard to control and consume most of the individual’s cognitive resources, leaving the person with less ability to perform everyday tasks [7]. In particular, sadness rumination refers to “repetitive thoughts concerning one’s present distress and the circumstances surrounding the sadness” [8] and is an important risk factor for depression [9]. Studies found that the uncontrollability of ruminative thinking is the main cognitive factor that maintains and deepens depression [10].

The concept of well-being describes the individual’s mental state and psychological functioning [11]. Well-being is a multi-faced construct with two components: subjective happiness and life satisfaction, meaning people’s cognitive and affective evaluations of their lives [12]. The relationship between rumination and well-being is described in the literature as a cycle of underdetermination: if the person ruminates sad thoughts daily, then he/she will feel less happy, and happier people will have less tendency to ruminate on sad subjects [13]. Also, in a recent study, O’Connor and his colleagues concluded that high levels of worry and rumination are associated with clinically significant levels of anxiety and a low level of well-being in UK adults during the COVID-19 pandemic [6].

In a study published in 2023, Petwal and collaborators studied rumination as a mediator in the relationship between self-compassion and the severity of anxiety, but their mediation model was not significant. The authors concluded that rumination, a transdiagnostic process identified across psychological disorders, is a significant predictor of anxiety severity [14]. This study was conducted within a specific cultural setting, which may influence the expression and perception of anxiety and related constructs. Investigating these relationships across different cultural contexts would provide a more comprehensive understanding. Also, the study highlights the significance of self-compassion-based interventions, but it does not assess their effectiveness in reducing rumination and anxiety symptoms. Future research should evaluate the impact of such interventions on these variables.

In a 2021 study, Levkzuk and collaborators examined how attachment anxiety and avoidance relate to various health symptoms, such as depressive and vegetative symptoms. They found that emotion regulation difficulties mediate the relationship between attachment styles and health symptoms, highlighting the role of physical manifestations in the anxiety–well-being link [15].

The present study examines the relationship between anxious thoughts and well-being, with physical symptoms and sadness rumination as mediators, in young people who suffer from anxiety. The hypotheses we started with were that psychological well-being will correlate negatively, statistically significant, both with anxious thoughts as well as with the somatic symptoms of anxiety and with rumination; the second hypothesis was that both somatic symptoms and the rumination of sadness along with anxious thoughts are predictors of psychological well-being; and the last hypothesis was that the physiological symptoms of anxiety and the rumination of sadness mediate the relationship between anxious thoughts and psychological well-being in the evaluated youth. In essence, we propose that (a) well-being is related to anxious thoughts, physical symptoms and sadness rumination, (b) physical symptoms and sadness rumination will reinforce the relationship between anxious thoughts and well-being, and (c) physical symptoms and sadness rumination will mediate the effect of anxious thoughts on student’s well-being.

## 2. Methods

### 2.1. Study Design

The research was conducted in several stages after we collected the data. Since this was a cross-sectional study, we began by analyzing the correlations between the variables investigated. Then, we performed multiple hierarchical regression, also analyzing the possibility of multicollinearity between the studied variables. In the final stage, we conducted the study of the mediation model proposed in the paper.

### 2.2. Setting

The participants were selected primarily based on the existence of an anxiety disorder, either in the past or present, this being the main selection criterion for the subjects. For this, when presenting the informed consent, we included an interview section that determined each candidate’s inclusion in the group. Additionally, the participants are students at the Polytechnic University of Bucharest, as we considered easier accessibility to the subjects we evaluated. Their assessment took place over a period from October 2023 to March 2024, with great attention paid to data collection.

### 2.3. Participants

Our data reflects a community sample of 198 participants in this study, chosen through stratified sampling as well as the criterion of participant availability, 94 males (47.47%), 104 females (52.53%), all of them with ages between 19 and 35 years, all of them having an anxiety disorder in their past, having medical treatment for this at a certain point, or being in psychotherapy. All participants are university students from different faculties who have been studying for different years. When selecting the subjects, a stratified sampling technique was used, dividing university students by faculty and then by specialization while also considering the availability of participants. These participants were students from the Faculty of Educational Sciences, Social Sciences, and Psychology. Another important factor in participant selection was their declaration of having experienced anxiety either in the past or at the time of the evaluation. The percentage of participants studying Psychology is 55%, Human Resources is 23.5%, and Social Work is 21.5%.

### 2.4. Variables and Data Sources

To measure rumination, we used the Sadness and Anger Rumination Questionnaire [9,16]. This instrument measures sadness rumination and anger rumination, as well as a general score for rumination. The questionnaire has 11 items for each type of rumination and two scales: anger rumination and sadness rumination. Participants answer on a 5-point scale (1 = never, 5 = always) [9]. Internal consistency in the current study was 0.93 for anger rumination and 0.89 for sadness rumination.

To measure the physical symptoms of anxiety, we used The Burns Anxiety Inventory (Burns). Specifically, the Burns-A is a self-report instrument developed with the aim of evaluating anxiety symptoms. Burns–A has 33 items that focus on three anxiety dimensions: anxious feelings, anxious thoughts, and physical symptoms. Participants answer on a 4-point scale (0 = never, 3 = always). Internal consistency in the current study was 0.71 for anxious feelings, 0.66 for anxious thoughts and 0.75 for physical symptoms.

To measure well-being, we used Ryff’s Psychological Well-Being Scale (PWB-R) [17]; adapted from [18]. The test has 42 items, with a total score of well-being and 6 scales: autonomy, control of situations and life, personal development, positive interpersonal relationships, clarity of goal, and self-acceptance. Participants answer on a 7-point scale (1 = strongly agree, 7 = strongly disagree). Internal consistency in the current study for the total score was 0.86.

### 2.5. Bias

Informed consent was obtained from all the student participants using an online form. Common ethical practices in research with human subjects were followed. The study was approved by the Ethics Committee of the university with which the investigators are associated. All the questionnaires were incorporated into a Google Form, and the form was shared by the first author via email and social media with a large number of students from different faculties and specializations. The questionnaires were carefully completed in full, and the main selection criterion for participants was based on the existence of an anxiety disorder, either in the past or present. Additionally, data were collected in a direct manner using pen and paper by the other authors. The data were completed both online and in paper form, and since we were proposing a path analysis model, we wanted to ensure that in the screening phase, we could control the investigated variables, so each questionnaire was numbered in advance. The questionnaires were administered by the authors based on certain variables designated before testing. Furthermore, during the questionnaire administration phase, we aimed to ensure that there was no missing data when completing the instruments.

### 2.6. Study Size

Since the subjects were selected based both on availability and the fact that we tried to completely control data collection, ensuring there were no missing data and that we had access to the selected subjects throughout the entire duration of the study, the total number of participants was the one mentioned earlier in the paper.

### 2.7. Quantitative Variables

For measuring the studied variables, we chose instruments that are reliable with respect to the Romanian population, as mentioned earlier.

### 2.8. Statistical Methods

As statistical-mathematical analysis methods, we used the analysis of descriptive statistical indicators of variability and the shape of the data distribution curve, Pearson correlation, partial correlation, hierarchical multiple regression, *T*-test, variance inflation factor (VIF), and the path analysis model 6 by Macro Process.

## 3. Results

Our first hypothesis was that there are strong correlations between our studied variables. Results show that well-being is significantly correlated with anxious thoughts (r = −0.74, *p* < 0.01), physical symptoms (r = −0.82, *p* < 0.01) and sadness rumination (r = −0.82, *p* < 0.01). The Pearson correlation values indicate a strong negative linear association with high and inversely proportional intensity, statistically significant, between the normally distributed variables (Skewness and Kurtosis within ±1), meaning that a lower level of the variables anxious thoughts, physical symptoms, and sadness rumination is associated with an increase in the level of well-being, and 54.76% of the variance of the well-being variable is explained by anxious thoughts, while 67.24% of the well-being variable is explained by physical symptoms and sadness rumination. The correlations analyzed are significant at a significance threshold of *p* < 0.05 (*) and *p* < 0.01 (**), which explains the probability of making a mistake by omission or chance and highlights the variation link between the analyzed variables or the confidence interval within which the values of the variables become significant.

These findings support the fact that student’s well-being is significantly related to their anxious thoughts, physical symptoms, and sadness rumination. We aim to identify whether there is a valid predictive model in which the level of well-being can be predicted by the variables anxious thoughts, physical symptoms, and sadness rumination and also if there are variables that influence the association between them in the form of mediation.

### 3.1. The Incremental Validity of Physical Symptoms and Sadness Rumination

The second hypothesis was that well-being is predicted by anxious thoughts, physical symptoms and sadness rumination. Table 1 presents the results of the hierarchical multiple regression analysis predicting well-being from anxious thoughts (Step 1), from anxious thoughts and physical symptoms (Step 2), and from anxious thoughts, physical symptoms and sadness rumination (Step 3). The results showed that: in the first step of regression, anxious thoughts contributed significantly to the regression model (*F*(1; 197) = 243.62, *p* = 0.001) and accounted for about 55% of the variation in well-being; in the second step of regression anxious thoughts and physical symptoms contributed significantly to the regression model (*F*(2; 197) = 262.43, *p* = 0.001) and accounted 72% of the variation in well-being, underlining the superiority of the second model by comparison with the first model; sadness rumination introduced in the third step of regression next to anxious thoughts and physical symptoms contributed significantly to the regression model (*F*(3; 197) = 180.41, *p* = 0.00), and accounted 73% of the variation in well-being, underlining the superiority of the model with three predictors.

Looking at the unique individuals’ contributions of the predictors, the result (Table 1) and (Table 2) shows that anxious thoughts (*β* = −0.74, *t* = −15.60, *p* < 0.01), physical symptoms (*β* = −0.61, *t* = −11.22, *p* < 0.001) and sadness rumination (*β* = −0.26, *t* = −2.27, *p* < 0.001) significantly influence well-being. Furthermore, we may observe that all the predictors’ variables have a negative value, which shows us that a higher level of well-being is better explained by the decreasing levels of the variable’s anxious thoughts and physical symptoms, as well as by the decreasing of sadness rumination, though sadness rumination has a lower value of prediction. In conclusion, the second hypothesis is confirmed.

Analyzing the values of the indicators of the distribution curve shape for each evaluated variable, we cannot say that the obtained distributions fall into cases of pronounced skewness or kurtosis value of the partial correlation coefficients presented in Table 3: the standardized correlation value for the well-being prediction equation based on the variable anxious thoughts is r = −0.74 and for physical symptoms is r = −0.83, which signifies that 54.76% of the variance of the well-being and anxious thoughts variables is common, respectively, 68.89% of the variance of the well-being and physical symptoms variables is common; the partial correlation value between the well-being and anxious thoughts variables, controlling for the influence of other variables, is r = −0.38, the partial correlation value between the well-being and physical symptoms variables, controlling for the influence of other variables, is r = −0.63; the partial correlation value between the well-being and anxious thoughts variables without the control of other variables is r = −0.21, the partial correlation value between the well-being and physical symptoms variables without the control of other variables is r = −0.42, values significant at *p* < 0.001.

On the other hand, the standardized correlation value for the well-being prediction equation based on the variables anxious thoughts is r = −0.74 and for sadness rumination is r = −0.82, which signifies that 54.76% of the variance between the well-being and anxious thoughts variables is common, and 67.24% of the variance between the well-being and sadness rumination variables is common; also, the partial correlation value between the well-being and anxious thoughts variables, controlling for the influence of other variables, is r = −0.40, and the partial correlation value between the well-being and sadness rumination variables, controlling for the influence of other variables, is r = −0.62; the partial correlation value between the well-being and anxious thoughts variables without the control of other variables is r = −0.23, and the partial correlation value between the well-being and sadness rumination variables without the control of other variables is r = −0.41, values significant at *p* < 0.001.

Although the value of the mentioned correlations is high, negative, and suggests the existence of collinearity, the inclusion in the well-being prediction model of the variables anxious thoughts, physical symptoms, and sadness rumination brings significant information without overestimating the determination coefficient, without changing the variation of the already analyzed estimated coefficients, and without distorting the interpretation of the model, maintaining the confidence intervals: CI for the *β* coefficients of the variables anxious thoughts (−0.95; −0.46) and physical symptoms (−1.20; −0.84) for the well-being prediction model according to the variables anxious thoughts and physical symptoms with *t* = −11.22 at *p* = 0.0001, respectively CI for the average of the variables anxious thoughts (−1; −0.51) and sadness rumination (−0.72; −0.50) for the well-being prediction model according to the variables anxious thoughts and sadness rumination with *t* = −10.89 at *p* = 0.0001, underscores the significant value of the variables introduced in the prediction model.

The coefficient of determination for the equation predicting the level of well-being based on the variables anxious thoughts and physical symptoms is ΔR^2^ = 0.73 with *F*(2; 197) = 262.43 at *p* < 0.001, and the coefficient of determination for the equation predicting the level of well-being based on the variables anxious thoughts and sadness rumination is ΔR^2^ = 0.72 with *F*(2; 197) = 254.36 at *p* < 0.001, values that suggest the variables physical symptoms and sadness rumination have a positive influence on the relationship between the well-being and anxious thoughts variables.

These findings are supported by the Student’s *t*-test value: by introducing the variable anxious thoughts into the first well-being prediction model *t* = −5.71 > −2 at *p* < 0.001 and by introducing the physical symptoms variable *t* = −11.22 > −2 at *p* < 0.001; by introducing the variable anxious thoughts into the second well-being prediction model *t* = −6.15 > −2 at *p* < 0.001 and by introducing the sadness rumination variable *t* = −10.89 > −2 at *p* < 0.001; the Student’s *t*-test values highlight the fact that introducing the variables physical symptoms and sadness rumination into the well-being prediction models significantly increases the value of the regression model, information that leads to the rejection of the null hypothesis and allows the acceptance of the hypothesis according to which the variables physical symptoms and sadness rumination strongly and positively influence the relationship between the well-being and anxious thoughts variables. Another important indicator is the value of the standard residual for the first prediction model, which is within the interval (−1.76; 2.54) and for the second prediction model being within the interval (−2.08; 2.70), an indicator that suggests the regression models predict the state of well-being quite well, with no aberrant data.

To determine the presumed multicollinearity due to the increased importance of the variable’s physical symptoms and sadness rumination introduced in the prediction models, we analyzed the correlation matrix between the variables included in the prediction equations. As can be seen in the analysis already described, the correlation values between the variables well-being and anxious thoughts, respectively physical symptoms, are negative and high in intensity, indicating an inversely proportional relationship, while the correlation value between the variables anxious thoughts and physical symptoms is positive and high, suggesting a directly proportional relationship (r = 0.72, *p* < 0.001).

Also, the correlation values between the variables well-being and anxious thoughts, respectively sadness rumination, are negative and high in intensity, indicating an inversely proportional relationship, while the correlation value between the variables anxious thoughts and sadness rumination is positive and high, suggesting a directly proportional relationship (r = 0.71, *p* < 0.001). This aspect suggests checking for the presence of collinearity between the variables anxious thoughts and physical symptoms, respectively, anxious thoughts and sadness rumination.

Therefore, analyzing the covariance value between the variables anxious thoughts and physical symptoms, which is equal to −0.008, respectively anxious thoughts and sadness rumination, which is equal to −0.005, we observe that each of the two variables within the two regression models records a negative value, very low <−1, which indicates a weak relationship and suggests the absence of collinearity. This fact is supported by the value of the determinant > 0, the analysis of the tolerance value and the inflation factor of tolerance: in the case of the first regression model, the value τ > 0.1 corresponding to a determination coefficient ΔR^2^ < 1, therefore a weaker linear link between the independent variables anxious thoughts and physical symptoms (τ = 0.48), respectively anxious thoughts and sadness rumination (τ = 0.49).

The tolerance inflation factor VIF for the variables anxious thoughts and physical symptoms = 2.09, and VIF for the variables anxious thoughts and sadness rumination = 2.01. All these analyzed data suggest the absence of collinearity and the necessity to verify the mediation effect of the variables.

### 3.2. Indirect Effects of Anxious Thoughts on Well-Being

Our last hypothesis was that there is a serial mediating effect of physical symptoms and sadness rumination on the relationship between anxious thoughts and well-being.

The PROCESS macro program was used to test the serial mediation model 6 (Hayes, 2013, https://psycnet.apa.org/record/2013-21121-000 (accessed on 5 October 2024)); Table 4 shows the serial mediating effects of physical symptoms and sadness rumination on the relationship between anxious thoughts and well-being. As shown in Table 4, 95% confidence intervals for all indirect effects do not include zero and all indirect effects are significant. Physical symptoms mediate the effect of anxious thoughts on well-being (*β* = 0.98, *SE* = 0.06, *p* < 0.001). Sadness rumination mediates the effect of anxious thoughts on well-being (*β* = 0.12, *SE* = 0.07, *p* < 0.001). Physical symptoms and sadness rumination serially mediate the effect of anxious thoughts on well-being (*β* =−0.17, *SE* = 0.07, *p* < 0.001). As seen in Table 4, the direct effect of anxious thoughts on well-being is −0.67 (*SE* = 0.12, *p* < 0.001), and its total indirect effect is −0.39 (*SE* = 0.15, *p* < 0.001) (Figure 1).

## 4. Discussion

The present study examines the relationship between anxious thoughts and well-being, with physical symptoms and sadness rumination as mediators, in young people who suffer from anxiety. In essence, we propose that (a) well-being is related to anxious thoughts, physical symptoms and sadness rumination, (b) physical symptoms and sadness rumination will reinforce the relationship between anxious thoughts and well-being, and (c) physical symptoms and sadness rumination will mediate the effect of anxious thoughts on student’s well-being. The findings are consistent with previous research. People who experienced anxiety in the past have an autobiographic memory bias toward recalling more negative anxiety-related memories [19], but also a high tendency to think of possible negative future events that they strongly believe will occur [20], and this tendency will affect their well-being. Niziurski and Shaper conducted a study of 904 participants’ well-being in Germany and the US during COVID-19 restrictions and concluded that their current psychological well-being related to how they remembered events and thought of their future [21]. Karabati and his colleagues concluded that rumination leads to subjective unhappiness and diminishes life satisfaction in white-collar employees from the US and Turkey [13].

Mediation Model 6 validates the fact that there is a mediating relationship in the sense that people with anxious thoughts that produce strong physical symptoms and sadness rumination decrease the level of well-being or physical symptoms and sadness rumination mediate the relationship between anxious thoughts and well-being.

All these analyses have validated the proposed hypotheses, namely the existence of the relationship between anxious thoughts and well-being, with physical symptoms and sadness rumination as mediators, in young people who suffer from anxiety. Our study has two main limitations. First, the relatively modest sample size (198 participants) and the limited geographical distribution of the subjects since they were all students at the same university. The second limitation is the cross-sectional nature of our study, which makes a robust investigation of causal effects impossible. Another limitation of the study could be considered the stage in which we could have conducted the structural equation modeling (SEM) of the proposed model, which we have in mind for future, more extensive studies to more precisely control both endogenous and exogenous variables that might interfere. A larger scale study could lead to national strategic proposals in attending to the mental health of university students and improving their well-being. The purpose of this study was to examine the relationship between anxious thoughts and well-being, with physical symptoms and sadness rumination as mediators, in young people who suffer from anxiety.

Starting from the specialized literature which describes anxiety as an emotional problem manifested through worry (the fear for future events) and rumination (the cognitive representation of past stressful events as ‘repetitive thoughts concerning one’s present distress and the circumstances surrounding the sadness’ [8], p. 404, with strong consequences on health and well-being, having a prevalence of 3.6% [1], we verified on a sample of 198 students from different faculties, in different years of study, 47.47% males, 52.53% females, all between the ages of 19–35 years, all of them having had an anxiety disorder in their past, or in present:(a)Well-being is related to anxious thoughts, physical symptoms, and sadness rumination;(b)Physical symptoms and sadness rumination will reinforce the relationship between anxious thoughts and well-being;(c)Physical symptoms and sadness rumination will mediate the effect of anxious thoughts on student’s well-being.

Starting from these ideas, we used the Sadness and Anger Rumination Questionnaire to measure rumination [9]. To measure the physical symptoms of anxiety, we used The Burns Anxiety Inventory (Burns). To measure well-being, we used Ryff’s Psychological Well-Being Scale (PWB-R) [17], adapted from [18].

The results highlighted the fact that well-being is significantly negatively correlated with anxious thoughts, physical symptoms, and sadness rumination, variables that are normally distributed with high percentages of the well-being variable being explained by them. These initial findings validated the fact that the student’s well-being is significantly related to their anxious thoughts, physical symptoms, and sadness rumination.

Starting from these data, in order to verify if (b) physical symptoms and sadness rumination will reinforce the relationship between anxious thoughts and well-being, we analyzed valid prediction models in which the level of well-being can be predicted by the variables anxious thoughts, physical symptoms, and sadness rumination.

All these analyses have validated the proposed hypotheses, namely the existence of the relationship between anxious thoughts and well-being, with physical symptoms and sadness rumination as mediators, in young people who suffer from anxiety.

Our study’s findings highlight the importance of addressing both cognitive and physical aspects of anxiety to improve well-being. Interventions that target anxious thoughts, manage physical symptoms, and reduce sadness rumination may be particularly effective in enhancing mental health among young individuals, especially in the context of ongoing global health challenges.

In conclusion, the interplay between anxious thoughts, physical symptoms, and sadness rumination plays a critical role in determining well-being among young people with anxiety. The COVID-19 pandemic has intensified these issues, making it imperative to develop comprehensive mental health strategies that address both psychological and physical components of anxiety.

## 5. Conclusions

In summary, this study explored the complex relationship between anxious thoughts and well-being in young people who experience anxiety, with physical symptoms and sadness rumination acting as mediators. The findings highlight that well-being is closely tied to anxious thoughts, physical symptoms, and sadness rumination. Furthermore, these mediators appear to amplify the impact of anxious thoughts on well-being, reinforcing the notion that physical symptoms and sadness rumination play a significant role in shaping the mental health landscape for young individuals facing anxiety.

The implications of these findings are noteworthy for both mental health professionals and educators. Recognizing the mediating roles of physical symptoms and sadness rumination may assist practitioners in developing targeted interventions that address these factors to improve young people’s well-being. Additionally, these insights could inform preventive strategies in educational and community settings to mitigate the impact of anxiety on well-being through proactive support.

However, several limitations must be considered. The sample may limit generalizability, as it predominantly includes young individuals with anxiety, potentially narrowing the scope of applicability to other age groups or broader populations. Additionally, self-reporting methods may introduce bias, affecting the reliability of data on internal states such as anxious thoughts and sadness rumination. These limitations underscore the need for caution in generalizing findings beyond the study’s sample and methodology.

Future research should seek to address these limitations by exploring diverse age groups and employing longitudinal designs to examine changes over time. Further investigation into other potential mediators or moderators, such as social support or coping strategies, may also deepen our understanding of the relationship between anxious thoughts and well-being. Continued research in this area will provide a more nuanced understanding, potentially informing comprehensive interventions to improve mental health outcomes for young people experiencing anxiety.

## Figures and Tables

**Figure 1 behavsci-14-01109-f001:**
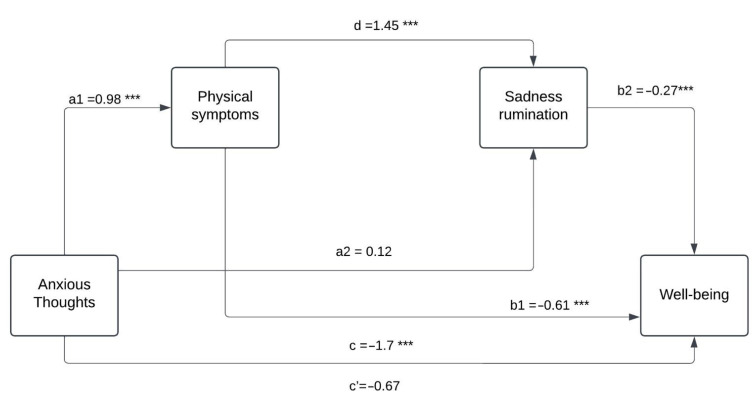
Path analysis diagram showing the mediational relationship of physical symptoms and sadness rumination mediating the effect of anxiety thoughts on Well-being. *** *p* < 0.001.

**Table 1 behavsci-14-01109-t001:** Summary of hierarchical multiple regression analysis predicting well-being from anxious thoughts (Step 1), from anxious thoughts and physical symptoms (Step 2), and from anxious thoughts, physical symptoms and sadness rumination (Step 3).

Step	Predictor	R^2^	ΔR^2^	*β*	*F*
1	Anxious Thoughts	0.55	0.55 ***	−0.74 ***	243.62
2	Anxious Thoughts	0.73	0.18 ***	−0.31 ***	262.43
	Physical Symptoms			−0.61 ***	
3	Anxious Thoughts	0.74	0.01 **	−0.29 ***	180.41
	Physical Symptoms			−0.37 ***	
	Sadness Rumination			−0.26 ***	

** *p* < 0.01; *** *p* < 0.001. Each step adds predictors to the model, and the change in R^2^ (ΔR^2^) reflects the additional variance explained by each step. R^2^ indicates the proportion of variance in well-being explained by the model at each step, while *F* is the test statistic for each step’s model fit.

**Table 2 behavsci-14-01109-t002:** Central tendency values for the predictors of the regression equation physical symptoms, sadness rumination, anxious thoughts—mean, standard deviation, estimation error, Skewness, Kurtosis, confidence intervals, Student’s *t*-test (N = 198).

Predictor	M	SD	Skewness	Kurtosis	*SE*	*p*	95% CI	*t*	CV
Anxious Thoughts	17.54	4.76	−0.22	−0.25	0.11	<0.01	[−1.91, −1.49]	−15.61	27%
Physical Symptoms	26.30	6.49	−0.51	−0.34	0.91	<0.001	[−1.20, −0.84]	−11.22	25%
Sadness Rumination	36.01	10.44	−0.95	−0.22	0.12	<0.001	[−0.51, −0.03]	−2.27	29%
Well-being	101.64	10.96	−0.30	−1.25					10%
Residual	0	0.99			0	<0.001	[−1.76, 2.54]		

M = Mean, SD = Standard Deviation, *SE* = Standard Error, CI = Confidence Interval, *t* = Student’s *t*-test value, CV = Coefficient of Variation. Confidence intervals are presented in square brackets. Values for skewness and kurtosis reflect the distribution characteristics of each predictor.

**Table 3 behavsci-14-01109-t003:** Values of the predictors of the regression equation physical symptoms, sadness rumination, anxious thoughts—Beta coefficients, standard deviation, estimation error, significance threshold, confidence intervals, Student’s *t*-test, determination coefficients, correlation coefficients (N = 198).

Predictors Model 1	Well-Being													
a_1_ = 140.92	*β*	*SE*	*p*	95% CI	*t*	*F*	R^2^	R Zero-Order	R Partial	R Part	Tolerance	VIF	Cov.	Estimated Standard Error
Anxious thoughts	−0.71	0.12	<0.001	[−0.95, −0.46]	−5.71	243.62	0.55	−0.74	−0.38	−0.21	0.48	2.09	0.15	7.34
Physical symptoms	−1.02	0.09	<0.001	[−1.20, −0.84]	−11.22	262.44	0.73	−0.83	−0.63	−0.42	0.48	2.09	−0.008	5.73
residual (−1,76; 2,54).														
**Predictors Model 2**	**Well-Being**													
**a_2_ = 137.01**	** *β* **	** *SE* **	** *p* **	**95% CI**	** *t* **	** *F* **	**R^2^**	**R Zero-Order**	**R Partial**	**R Part**	**Tolerance**	**VIF**	**Cov.**	**Estimated Standard Error**
Anxious thoughts	−0.76	0.12	<0.001	[−1, −0.51]	−6.15	243.62	0.55	−0.74	−0.40	−0.23	0.49	2.01	15	7.34
Sadness rumination	−0.61	0.06	<0.001	[−0.72, −0.50]	−10.89	254.36	0.72	−0.82	−0.62	−0.41	0.49	2.01	−0.005	5.80
residual (−2,08; 2,70).														

**Table 4 behavsci-14-01109-t004:** Serial mediation: Indirect effects of Anxious thoughts on Well-being (N = 198). Serial Mediation Model: Indirect Effects of Physical Symptoms and Sadness Rumination on the Relationship Between Anxious Thoughts and Well-being (N = 198).

Predictor	Dependent Variable	*β*	*SE*	*p*	95% CI
Direct Paths					
Anxious Thoughts	Physical Symptoms	0.98	0.06	<0.001	[0.85, 1.17]
Physical Symptoms	Sadness Rumination	1.45	0.05	<0.001	[1.30, 1.60]
Sadness Rumination	Well-being	−0.27	0.12	<0.001	[−0.05, −0.03]
Anxious Thoughts	Well-being (Direct)	−0.67	0.12	<0.001	[−0.91, −0.43]
Indirect effects					
**Path**	**Indirect Effect (*β*)**	** *SE* **	** *p* **	**Bootstrapped 95% CI**	
Anxious Thoughts → Physical Symptoms → Well-being	−0.61	0.05	<0.001	[−1.01, −0.02]	
Anxious Thoughts → Physical Symptoms → Sadness Rumination → Well-being	−0.17	0.07	<0.001	[−0.07, −0.09]	
Total Indirect Effect	−0.39	0.15	<0.001	[−0.07, −0.09]	
Total and model fit					
**Effect Type**	** *β* **	** *SE* **	** *p* **	**95% CI**	
Total Effect (X → Y)	−1.71	0.11	<0.001	[−1.4, −0.74]	
R^2^ for Physical Symptoms	0.52		<0.001		
R^2^ for Sadness Rumination	0.89		<0.001		
R^2^ for Well-being	0.73		<0.001		
*F* for Physical Symptoms	213.61				
*F* for Sadness Rumination	199.71				
*F* for Well-being	180.41				

X = Anxious Thoughts, M1 = Physical Symptoms, M2 = Sadness Rumination, Y = Well-being; X = Anxious Thoughts, M1 = Physical Symptoms, M2 = Sadness Rumination, Y = Well-being; All paths, direct effects, and indirect effects are represented with beta coefficients (*β*), standard errors (*SE*), *p*-values, and 95% confidence intervals (CI). Confidence intervals for indirect effects were obtained using 5000 bootstrapped samples.

## Data Availability

The data presented in this study are available on request from the corresponding author.

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
