# Peer review of "Anxious Thoughts and Well-Being in University Students with Anxiety in the Two Years After the COVID-19 Pandemic: The Mediational Relationship of Physical Symptoms and Sadness Rumination"

_behavsci, 2024, doi:10.3390/bs14111109_

Round 1
Reviewer 1 Report
Comments and Suggestions for Authors
The study examines the relationship between anxious thoughts and well-being in young people with anxiety, focusing on the roles of physical symptoms and rumination on sadness as mediators. The research is conducted within the first two years following the COVID-19 pandemic, making it a timely and relevant topic. Here are my comments regarding this study:
Abstract:
In the abstract, it would be beneficial to mention the statistical method used, specifically hierarchical regression.
Introduction:
I couldn’t find a strong rationale for why the author chose physical symptoms and sadness rumination as mediators in the relationship between anxious thoughts and well-being.
It would be helpful to include a section that covers the study’s background and clearly outlines the research hypothesis.
Additionally, there is a lack of strong evidence regarding gaps in previous studies, particularly concerning the mediating roles of physical symptoms and sadness rumination.
Method:
There is a lack of strong evidence that authors were used “multiple hierarchical regression” for their study.
Authors mentioned “The participants were selected primarily based on the existence of an anxiety disorder, either in the past or present”. How did they measure it?
Need more explanation how the authors did approach to participant. I just saw one university name: Polytechnic University of Bucharest.
The authors mentioned that they have used “stratified sampling”. How did they define strata in their research. Need more details.
Authors mentioned “All participants are university students at different faculties, in different years of study”. Name of university/es and number/ percentage of participants for every university needed.
Sample size is not well presented. How did authors find sample size?
Discussion:
The discussion part is not strong enough I can see that the authors were referenced 2015, 2014, 2019 which is about before covid-19. However, the study is about anxiety in the two years after the COVID 19 pandemics. The only current references in discussion (Niziurski & Shaper, 2021)
Author Response
Abstract – In the abstract, it would be beneficial to mention the statistical method used, specifically hierarchical regression.
Thank you for your comment. We have added the statistical method.
Introduction
I couldn’t find a strong rationale for why the author chose physical symptoms and sadness rumination as mediators in the relationship between anxious thoughts and well-being.
Thank you for your comment. We have choosd physical symptoms and sadness rumination as mediators in the relationship between anxious thoughts and well-being because these factors can illustrate how anxiety impacts mental health through specific pathways. Physical symptoms and sadness rumination capture different but complementary pathways from anxiety to diminished well-being: the physiological and cognitive-emotional routes, respectively. Here’s why each is significant: Physical Symptoms: Anxiety often manifests physically, resulting in symptoms like tension, fatigue, headaches, and digestive issues. These physical symptoms can reduce overall well-being by contributing to distress, discomfort, and even disability, creating a direct pathway by which anxiety affects quality of life. Sadness Rumination: Rumination involves repeatedly focusing on negative emotions and thoughts, which can amplify feelings of sadness and hopelessness. When people experience anxious thoughts, they might ruminate on their worries, which can deepen their emotional distress and lead to lower well-being. Rumination is linked to prolonged negative mood states, making it a strong candidate for understanding how anxiety translates into decreased well-being.
It would be helpful to include a section that covers the study’s background and clearly outlines the research hypothesis.
Thank you for your suggestion. We have made the suggested correction.
Additionally, there is a lack of strong evidence regarding gaps in previous studies, particularly concerning the mediating roles of physical symptoms and sadness rumination.
Thank you for your comment. We have clarified this by adding a paragraph on page 2.
Methods
There is a lack of strong evidence that authors were used “multiple hierarchical regression” for their study.
Thank you for your comment. We have clarified this by replacing Table 4 with a more detailed one.
Authors mentioned “The participants were selected primarily based on the existence of an anxiety disorder, either in the past or present”. How did they measure it?
Thank you for your comment. A important factor in participant selection was their declaration of having experienced anxiety either in the past or at the time of the evaluation.
Need more explanation how the authors did approach to participant. I just saw one university name: Polytechnic University of Bucharest.
Thank you for your comment. When selecting the subjects, a stratified sampling technique was used, dividing university students by faculty and then by specialization, while also considering the availability of participants. These participants were students from the Faculty of Educational Sciences, Social Sciences, and Psychology. Another important factor in participant selection was their declaration of having experienced anxiety either in the past or at the time of the evaluation. The percentage of participants studying Psychology is 55%, Human Resources is 23.5%, and Social Work is 21.5%.
All the participants were students at the Polytechnic University of Bucharest.
The authors mentioned that they have used “stratified sampling”. How did they define strata in their research. Need more details.
Thank you for your comment. See above answer. We have added this in the text, at page 3.
Authors mentioned “All participants are university students at different faculties, in different years of study”. Name of university/es and number/ percentage of participants for every university needed.
Thank you for your comment. See above answer. We have added this in the text, at page 3.
Sample size is not well presented. How did authors find sample size?
Thank you for your comment. See above answer. We have added this in the text, at page 3.
Discussion
The discussion part is not strong enough I can see that the authors were referenced 2015, 2014, 2019 which is about before covid-19. However, the study is about anxiety in the two years after the COVID 19 pandemics. The only current references in discussion (Niziurski & Shaper, 2021)
Thank you for your suggestion. We could not find more specific studies close to our model during and after the COVID 19 pandemics.

Reviewer 2 Report
Comments and Suggestions for Authors
In your model, the element labeled "Physical symptoms" should be revised. The method you're using requires the elements to be measured with ordinal data, not nominal data. For example, height is ordinal data—150 cm is clearly taller than 135 cm, and 135 cm is taller than 120 cm. In contrast, gender is nominal data; representing female with 1 does not imply inferiority to representing male with 2. Among your variables, "anxious thoughts" is measured with ordinal data, as a score of 5 reflects higher anxiety than a score of 1. However, the way you measure "Physical symptoms" does not follow the ordinal structure and needs to be corrected.
The citation format does not conform to the journal’s guidelines. Please revise it accordingly.
Additionally, please include a conclusion section, addressing the following four aspects: summary, implications, limitations, and future research directions.
In Figure 1, the size of the boxes is inconsistent, the alignment is uneven, and the arrow starting points are not uniform. Please adjust these elements to improve the overall presentation and ensure visual alignment.
Please review the consistency between the table headers and content. For instance, in Table 4, "total effect" should not be categorized alongside predictors like "anxious thoughts." This requires revision.
Table 2 lacks a header. What is the appropriate header for the "Anxious thoughts" column?
The headers in Table 1 are also inappropriate. If the content includes "Step 1, anxious thoughts," the header should not be "Predictor."
Ensure consistency in the table headers—are they labeled as "Predictor" or "Predictors"? Please maintain uniformity throughout.
In Table 1, some data is aligned to the top, while other data is aligned to the bottom. Please ensure consistent alignment across all rows.
Between lines 164 and 178, several percentages have an extra space (e.g., "55 %" instead of "55%"). I recommend removing the extra spaces. Interestingly, line 152 does not have this issue. Please revise accordingly.
The title seems somewhat lengthy; consider shortening it. This is just a suggestion, not a requirement.
Lastly, I suggest changing ".93" in the text to "0.93." This is optional but recommended for consistency.
Author Response
In your model, the element labeled "Physical symptoms" should be revised. The method you're using requires the elements to be measured with ordinal data, not nominal data. For example, height is ordinal data—150 cm is clearly taller than 135 cm, and 135 cm is taller than 120 cm. In contrast, gender is nominal data; representing female with 1 does not imply inferiority to representing male with 2. Among your variables, "anxious thoughts" is measured with ordinal data, as a score of 5 reflects higher anxiety than a score of 1. However, the way you measure "Physical symptoms" does not follow the ordinal structure and needs to be corrected.
Thank you for your suggestion.
Both physical symptoms and anxious thoughts are measured by subscales of The Burns Anxiety Inventory, and both are ordinal data. They both are measured on a 4-points Lickert scale (0 = never, 3 = always).Anxiety often manifests physically, resulting in symptoms like tension, fatigue, headaches, and digestive issues. Therefore, we consider "Physical symptoms" the correct element in our study.
The citation format does not conform to the journal’s guidelines. Please revise it accordingly.
Thank you for your suggestion. We made the suggested corrections.
Additionally, please include a conclusion section, addressing the following four aspects: summary, implications, limitations, and future research directions.
Thank you for your suggestion. We added this section.
In Figure 1, the size of the boxes is inconsistent, the alignment is uneven, and the arrow starting points are not uniform. Please adjust these elements to improve the overall presentation and ensure visual alignment.
Thank you for your suggestion. We made the suggested corrections.
Please review the consistency between the table headers and content. For instance, in Table 4, "total effect" should not be categorized alongside predictors like "anxious thoughts." This requires revision.
Thank you for your suggestion. We have modified Table 4.
Table 2 lacks a header. What is the appropriate header for the "Anxious thoughts" column?
Thank you for your suggestion. We have modified Table 2.
The headers in Table 1 are also inappropriate. If the content includes "Step 1, anxious thoughts," the header should not be "Predictor."
Thank you for your suggestion. We have modified Table 1.
Ensure consistency in the table headers—are they labeled as "Predictor" or "Predictors"? Please maintain uniformity throughout.In Table 1, some data is aligned to the top, while other data is aligned to the bottom. Please ensure consistent alignment across all rows.
Thank you for your suggestion. We have modified Table1, 2 and 4.
Between lines 164 and 178, several percentages have an extra space (e.g., "55 %" instead of "55%"). I recommend removing the extra spaces. Interestingly, line 152 does not have this issue. Please revise accordingly.
Thank you for your suggestion. We have made the suggested correction.
The title seems somewhat lengthy; consider shortening it. This is just a suggestion, not a requirement.
Thank you for your comment. We wanted to include all the variables in the title.
Lastly, I suggest changing ".93" in the text to "0.93." This is optional but recommended for consistency.
Thank you for your suggestion. We have made the suggested correction.

Reviewer 3 Report
Comments and Suggestions for Authors
The article succinctly presents and contextualizes the topic of anxiety and wellbeing in university students within a solid theoretical and empirical framework, referencing established research on anxious thoughts, physical symptoms, and sadness rumination. The research design, methods, and hypotheses are clearly laid out, with a well-defined sample of 198 participants, appropriate measures, and statistical techniques such as hierarchical multiple regression and mediation analysis.
The discussion of findings is coherent, supported by statistical evidence, and aligns with existing literature, although further elaboration on the implications of the results would strengthen the argument. The results are clearly presented, particularly in the mediation analysis, which supports the hypotheses about the relationship between anxious thoughts, physical symptoms, and wellbeing.
The article is well-referenced, drawing from relevant and recent scholarship, although additional depth in discussing the study's limitations could enhance the conclusion. The quality of English is clear, although minor improvements in phrasing and flow could improve readability. Overall, the article offers a compelling contribution to understanding anxiety's impact on student wellbeing.
Author Response
The article succinctly presents and contextualizes the topic of anxiety and wellbeing in university students within a solid theoretical and empirical framework, referencing established research on anxious thoughts, physical symptoms, and sadness rumination. The research design, methods, and hypotheses are clearly laid out, with a well-defined sample of 198 participants, appropriate measures, and statistical techniques such as hierarchical multiple regression and mediation analysis.
The discussion of findings is coherent, supported by statistical evidence, and aligns with existing literature, although further elaboration on the implications of the results would strengthen the argument. The results are clearly presented, particularly in the mediation analysis, which supports the hypotheses about the relationship between anxious thoughts, physical symptoms, and wellbeing.
The article is well-referenced, drawing from relevant and recent scholarship, although additional depth in discussing the study's limitations could enhance the conclusion. The quality of English is clear, although minor improvements in phrasing and flow could improve readability. Overall, the article offers a compelling contribution to understanding anxiety's impact on student wellbeing.
Thank you for your comments!